# The relationship between serum eosinophil peroxidase and major basic protein levels in relation to severity and response to H1 antihistamines in chronic spontaneous Urticaria

Cuc Thi Kim Nguyen[1,2], Lan Thi Pham[1,2], My Huyen Le[1,2], Doanh Huu Le[1,2], Minh Nguyet Vu[1,2]*

1 Department of Dermatology, Ha Noi Medical University, Hanoi, Vietnam, 2 National Hospital of Dermatology and Venereology, Hanoi, Vietnam

* vunguyetminh@hmu.edu.vn

## Abstract

### Objective

In chronic spontaneous urticaria (CSU), the significance of serum Eosinophil Peroxidase (EPO) and Major Basic Protein (MBP) levels as indicators of disease severity and response to antihistamine treatment is currently inadequately understood. This study explores the correlation between serum EPO and MBP levels, the severity of the disease, and the efficacy of antihistamines in patients with CSU.

### Methods

A cross-sectional study involved 120 CSU patients alongside 30 healthy controls. In addition, a cohort study targeted 60 patients diagnosed with severe CSU, defined by a Urticaria Activity Score over 7 days (UAS7 ≥ 28). Initially, these patients received a dose of 20 mg of bilastine, which could be increased to a maximum of 80 mg depending on the results of the Urticaria Control Test (UCT) conducted on days 15, 30, and 60. Baseline serum concentrations of EPO and MBP were assessed for all participants, with follow-up measurements conducted after two months for those with severe CSU utilizing an ELISA kit..

### Results

Serum EPO concentration in the severe CSU group was similar to that in the non-severe group (P = 0.33) and was higher than that in the healthy control group (P < 0.001). Serum MBP concentrations did not differ among these three groups (P = 0.19). Serum EPO and MBP concentrations did not correlate with UAS7 and UCT. They did not differ among antihistamine response groups in the severe CSU group (P > 0.05) and decreased after 2 months of antihistamine treatment (P < 0.05).

**Data availability statement:** Data cannot be publicly shared due to the regulatory restrictions imposed by the Institutional Review Board (IRB) at Hanoi Medical University, which are designed to safeguard personal data and confidential information. For legitimate inquiries, access to the data can be obtained through the IRB at Hanoi Medical University (contact: daihocyhn@hmu.edu.vn).

**Funding:** The author(s) received no specific funding for this work.

**Competing interests:** The authors have declared that no competing interest exist.

## Conclusions

Serum EPO and MBP levels are neither biomarkers predicting CSU severity nor factors predicting response to antihistamine in the severe group. This lack of association may help explain why treatments targeting eosinophil proliferation and chemotaxis have not been successful in clinical trials for patients with antihistamine-refractory CSU.

## Introduction

Chronic spontaneous urticaria (CSU) is a relatively common condition, affecting around 1% of the general population and significantly impacting patients' quality of life [1]. CSU is characterized by wheals and itching, angioedema, or both, passing nearly daily and lasting for more than six weeks without an identifiable trigger [2,3]. CSU's various clinical manifestations arise from its pathogenesis's complexity. Although mast cells are central to this condition, other immune cells, particularly eosinophils, also contribute to its effects [4,5]. Since 1980, the substantiation of eosinophilic infiltration in the skin lesions of CSU cases has been linked [6]. Peters et al. showed that nearly 50% of skin vivisection samples from CSU patients displayed eosinophil degranulation [7]. Interestingly, a significant difference was noted in the number of grains released from eosinophils at the edge of persistent papular lesions compared to non-persistent papular samples [8].

Eosinophil granules contain positively charged proteins that assist eosinophils in performing various physiological functions, similar to combating parasites and managing conditions like asthma, atopic dermatitis, bullous pemphigoid, and CSU [9,10]. Crucial eosinophil proteins include Major Basic Protein-1 and -2 (MBP-1 is the predominant form), Eosinophil Peroxidase (EPO), Eosinophil Cationic Protein (ECP), and Eosinophil-Derived Neurotoxin (EDN) [10]. These proteins detect mast cell degranulation through both IgE-independent pathways (via the Mas-related G protein-coupled receptor X2 (MRGPRX2)) [11]. And, in the case of EPO, through IgE-dependent pathways. Research has shown that patients with severe CSU have elevated levels of IgE anti-EPO in their blood compared to healthy individuals [12].

Given this environment, we question whether the concentrations of EPO and MBP could serve as implicit biomarkers for the activity level and treatment response to second-generation H1 antihistamines (sgAH1) in CSU. Only one small-sample study by Khanna et al. reported higher concentrations of MBP and EPO in a CSU group (comprising 11 cases) compared to a healthy person. This study did not identify a correlation between the serum concentrations of the two proteins and disease activity, as evaluated by the Urticaria Activity Score over a seven-day period (UAS7) [13].

We aimed to examine the relationship between serum concentrations of EPO and MBP and disease activity, and the efficacy of antihistamine treatment in patients with CSU.

## Materials and methods

### Study design

A cross-sectional study examined the relationship between serum EPO and MBP levels and the disease activity associated with CSU. A cohort study was also conducted to investigate the association between these two factors and the response to sgAH1. Both studies were carried out from March to September 2024 at the Chronic Urticaria and Urticaria Clinic in the Outpatient Department of the National Hospital of Dermatology and Venereology, Hanoi, Vietnam.

**IRB blessing status:** The Hanoi Medical University Institutional Ethical Review Board (HMU IRB) has approved the study protocol, assigned the reference number 1145/GCN-HMUIRB, dated December 28, 2023. The approval was signed by Professor Van Thanh Ta, Chairman of the Ethics Council. Written informed consent was obtained from all participants who were 18 years of age and older. For participants aged 16–18, consent was acquired from their guardians. This process was carried out prior to the collection of data and samples in the study. Each participant retains the right to withdraw from the survey without facing any adverse consequences.

### Study population

Our study encompassed 150 participants: 60 individuals diagnosed with severe CSU, 60 with non-severe CSU, and 30 healthy controls. Participants diagnosed with isolated CSU were recruited by the diagnostic criteria established by the EAACI/GA2LEN/EuroGuiDerm/APAAACI in 2022 [3]. All participants involved in the study were required to be at least 16 years of age and abstain from using sgAH1 for a minimum of 5 days before their enrollment. Additionally, they had not utilized any immunosuppressive treatments, such as systemic corticosteroids or methotrexate, for at least 1 month preceding the study. Furthermore, participants were instructed to avoid using any other medications, including nonsteroidal anti-inflammatory drugs (NSAIDs) and antibiotics, for at least 1 week before collecting their initial serum sample..

Exclusion criteria include: urticaria or angioedema resulting from urticarial vasculitis, urticaria pigmentosa, erythema multiforme, mastocytosis, hereditary angioedema, or drug-induced urticaria, chronic itching skin disorders, such as atopic dermatitis, bullous pemphigoid, dermatitis herpetiformis, senile pruritus, or psoriasis, and diseases that elevate serum levels of EPO and MBP were not included in the study (asthma [14], allergic rhinitis [15]) pregnant or nursing women.

A group of thirty healthy volunteers, carefully matched by age and sex, participated as control subjects in this study. These individuals were selected based on their absence of any history of inflammatory or allergic skin diseases, atopic conditions, infectious diseases, or other significant internal or surgical illnesses. Additionally, they had not utilized systemic antihistamines or immunosuppressive medications for the same length of time as the CSU patients before collecting and storing serum samples.

Patients underwent tests according to the EAACI/GA2LEN/EuroGuiDerm/APAAACI guidelines, which included a complete blood count (CBC), CRP, total IgE, and IgG anti-TPO [3]. The research protocol also administered autologous serum skin tests (ASST). Patients were instructed to assess their pruritus and wheal scores daily using a scale ranging from 0 to 3, facilitating a proactive approach to documenting their symptoms. The Urticaria Activity Score over 7 days (UAS7) was computed by aggregating these scores over one week. The patients diagnosed with CSU were classified into two distinct groups: those with severe CSU, characterized by a UAS7 score of 28 or greater, and those with non-severe CSU, indicated by a UAS7 score of less than 28 [16].

In the cohort of patients diagnosed with severe CSU, treatment was commenced with a standard dosage of bilastine (Bilaxten 20 mg/day), adhering to the guidelines set forth by EAACI, GA2LEN, EuroGuiDerm, and APAAACI. Follow-up assessments were carried out on days 15, 30, and 60 following the initiation of treatment. The efficacy of the treatment was evaluated through the Urticaria Control Test (UCT), which provided a framework for making any necessary adjustments to the bilastine dosage based on patient responses [3,17]. The bilastine dosage remained unchanged for patients categorized within the completely controlled group (UCT = 16) and the well-controlled group (UCT = 12–15). In contrast, for

patients identified as uncontrolled (UCT ≤ 11), the dosage was elevated by as much as four times, reaching a maximum of 80 mg per day [3]. A second serum sample was collected from the severe CSU group on day 60.

## Measurement of serum EPO and MBP levels

All serum samples were meticulously collected and stored at −80°C for two months. The evaluation of EPO and MBP levels was conducted using an ELISA kit (My BioSource, Inc., San Diego, CA, USA) within the Department of Hematology and Immunochemistry at the National Hospital of Dermatology and Venereology in Hanoi, Vietnam. These two assay kits are designed based on sandwich ELISA technology. A capture antibody, specifically the anti-EPO/MBP antibody, is pre-coated onto the bottom of each well. The detection antibody, which is biotin-conjugated anti-EPO/MBP, is crucial in identifying the target. The EPO/MBP antigen is sandwiched between the capture and detection antibodies. The concentration of EPO/MBP in the sample is directly proportional to the optical density (O.D.) absorbance measured at 450 nm. The concentration of EPO/MBP in a patient's serum sample is quantified by generating a standard curve. The serum samples were appropriately diluted to meet the required detection ranges: EPO (1.56–100 ng/mL) and MBP (3.12–200 ng/mL) to ensure precise measurements.

## Statistical analysis

The data were systematically entered into the REDCap platform at Vanderbilt University in Nashville, Tennessee, United States, and subsequently analyzed using Stata version 17.0, provided by StataCorp in College Station, Texas, United States. Quantitative variables are reported as the mean ± standard deviation when exhibiting a normal distribution, or as the median with interquartile range [Q1-Q3] when exhibiting a non-normal distribution. The normality of the distribution is evaluated using the Kolmogorov-Smirnov test for sample sizes exceeding 50, while the Shapiro-Wilk test is utilized for sample sizes below 50. Qualitative variables are expressed in terms of frequency and percentage. To assess differences among multiple comparisons, the Kruskal-Wallis test was applied, along with Dunn's corrections. Spearman's correlation was utilized to evaluate the relationships between EPO and MBP levels and various clinical and laboratory parameters. Furthermore, the Wilcoxon signed-rank test evaluated the differences in serum EPO and MBP concentrations before and after two months of treatment. A significance level of $P < 0.05$ was established for all analyses conducted.

## Results

### Characteristics of the study subjects

The study included 120 cases diagnosed with CSU and 30 healthy controls matched by age and sex. The average age of the healthy controls was 37.5 ± 12.7 years, with 63.3% being womanish. The group with severe CSU demonstrated higher CRP concentrations than the non-severe CSU group ($p = 0.02$). However, when a cut-off point of 5 mg/L was established, there was no significant difference in the rate of CRP elevation between the two groups. Furthermore, other clinical and laboratory parameters—including the duration of urticaria, positivity rates in ASST, incidence of combined angioedema, total IgE levels, IgG anti-TPO levels, eosinopenia, and basopenia—did not reveal significant differences between the groups (S1 Table).

### Correlation of serum EPO and MBP levels with disease severity

Serum EPO concentrations in the control group were substantially at the undetectable level (0 ng/mL), lower than those in the severe CSU (27.92 [21.18–36.89] ng/mL) and non-severe CSU (34.34 [24.01–43.61] ng/mL) ($P < 0.001$). No difference in serum EPO concentrations was observed between the severe and non-severe CSU groups ($P > 0.05$). Meanwhile, there was no difference in MBP concentrations among the three groups of severe CSU, non-severe CSU, and the control group ($P = 0.19$). Serum EPO and MBP concentrations do not correlate with UAS7 scores ($P > 0.05$) (Fig 1).

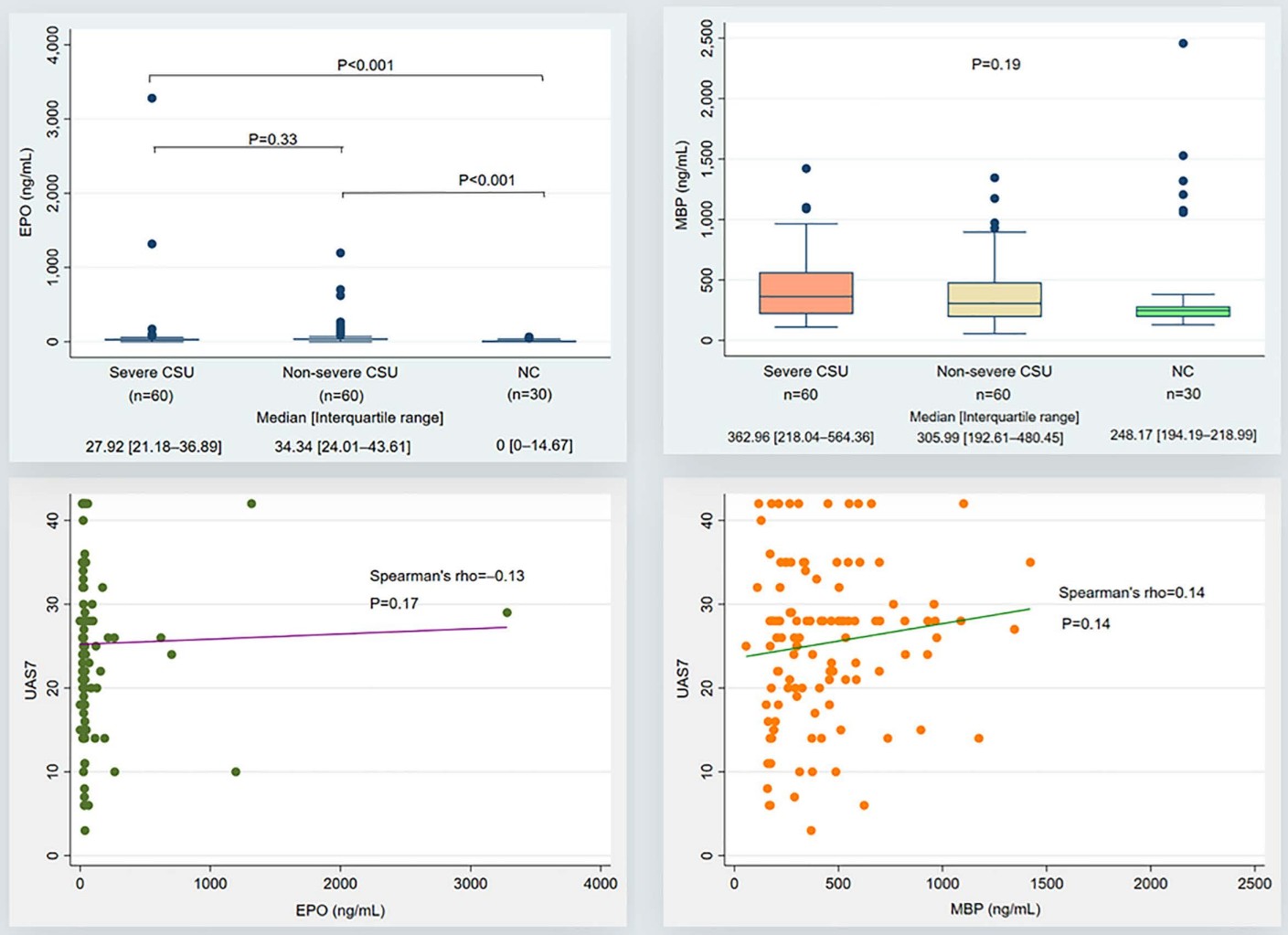

**Fig 1. Serum EPO and MBP concentrations between severe, non-severe CSU and controls (Kruskal-Wallis test with Dunn's multiple comparison test) and their association with UAS7 scores (Spearman correlation).** (CSU: Chronic spontaneous urticaria; EPO: Eosinophil Peroxidase; MBP: Major Basic Protein; ng/mL: nanogram/milliliter; NC: normal healthy controls; UAS7: Urticaria activity score over 7 days).

In CSU, serum EPO concentrations were weakly associated with age (Spearman's rho = −0.22; P < 0.05) but not with MBP (P > 0.05). These substances were weakly associated with IgG anti-TPO concentrations (P < 0.05). No correlation was found between EPO and MBP serum concentrations with disease duration, peripheral blood eosinophil counts, total IgE, and CRP levels. Additionally, no significant differences in serum EPO and MBP levels were observed when considering other characteristics, such as gender, presence of angioedema, and positive ASST (P > 0.05). (See S2 Table)

**The relationship between serum EPO and MBP levels and the response to antihistamines**

After 2 months of bilastine treatment, up to 35% of patients in the severe CSU group had uncontrolled symptoms despite taking up to 80 mg bilastine per day. EPO and MBP baseline concentrations were consistently comparable across the completely controlled, well-controlled, and uncontrolled groups, with no statistically significant differences observed (P > 0.05). The concentrations observed were not significantly related to the UCT scores (P > 0.05) (Fig 2).

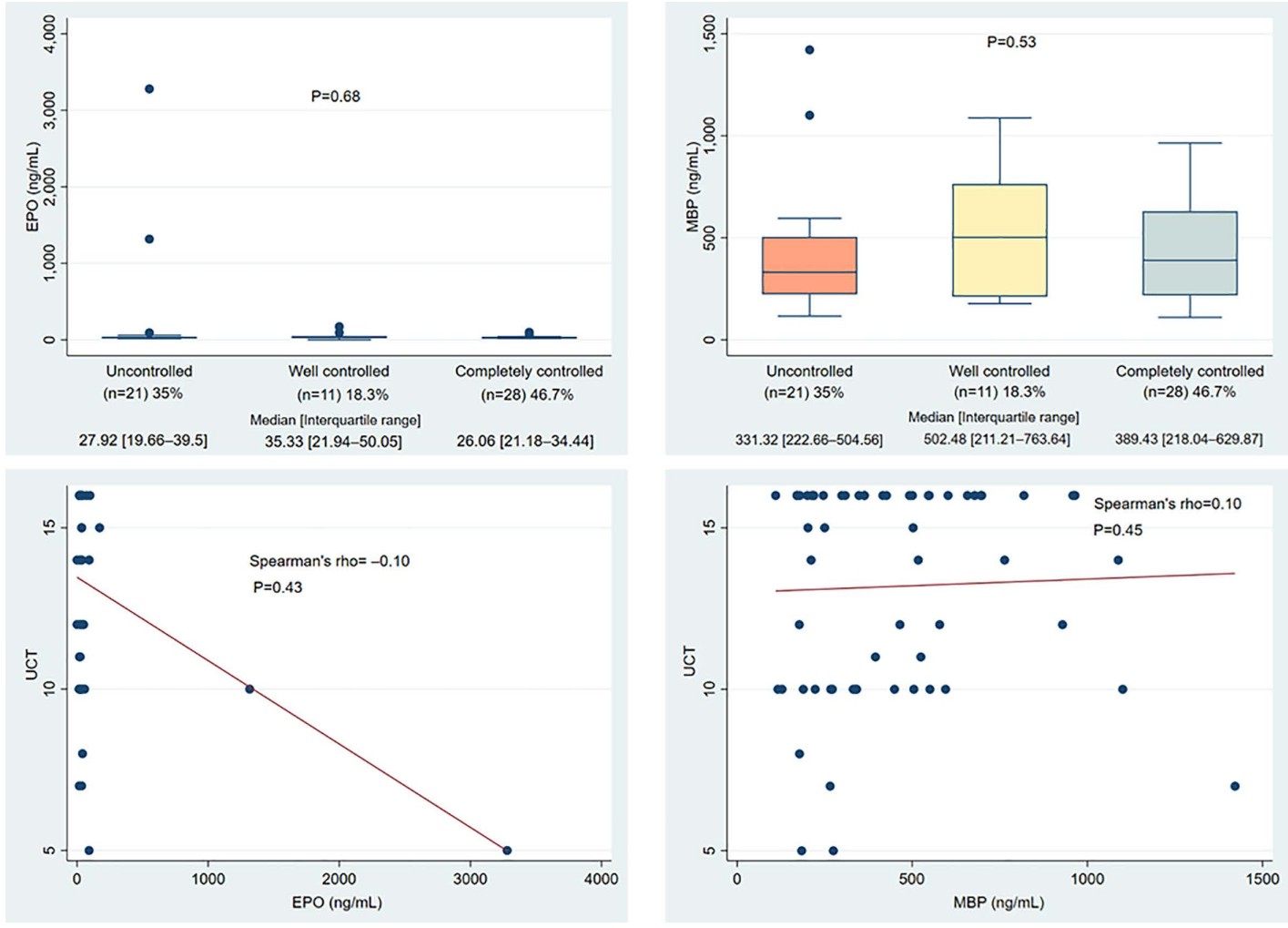

**Fig 2. Baseline serum EPO and MBP concentrations among treatment response groups to the antihistamine in the severe CSU (Kruskal-Wallis test) and their association with UCT scores (Spearman correlation).** (CSU: Chronic spontaneous urticaria; EPO: Eosinophil Peroxidase; MBP: Major Basic Protein; ng/mL: nanogram/milliliter; UCT: Urticaria Control Test).

We collected only 33 second-line serum samples. The serum EPO and MBP concentrations decreased after antihistamine treatment (P<0.05, Wilcoxon signed-rank test) (Fig 3).

## Discussion

EPO and MBP are both eosinophil granule proteins released when eosinophils degranulate. They partly represent the interaction between eosinophils and mast cells in CSU [5]. While EPO is only produced by eosinophils, MBP consists of MBP-1 (present in eosinophils and basophils but mostly in eosinophils) and MBP-2 (only in eosinophils) [10]. Our study showed that the serum EPO concentration of the patient group was significantly higher than that of the control group (P<0.001). In comparison, the serum MBP concentration of the patient group was higher than that of the control group, but this difference was not statistically significant (P>0.05). This result once again confirmed that eosinophil degranulation in CSU caused an increase in the concentration of these two proteins in the patient's serum. The study conducted by Khanna et al. observed that the concentration of MBP in the CSU group was significantly higher than in the control group,

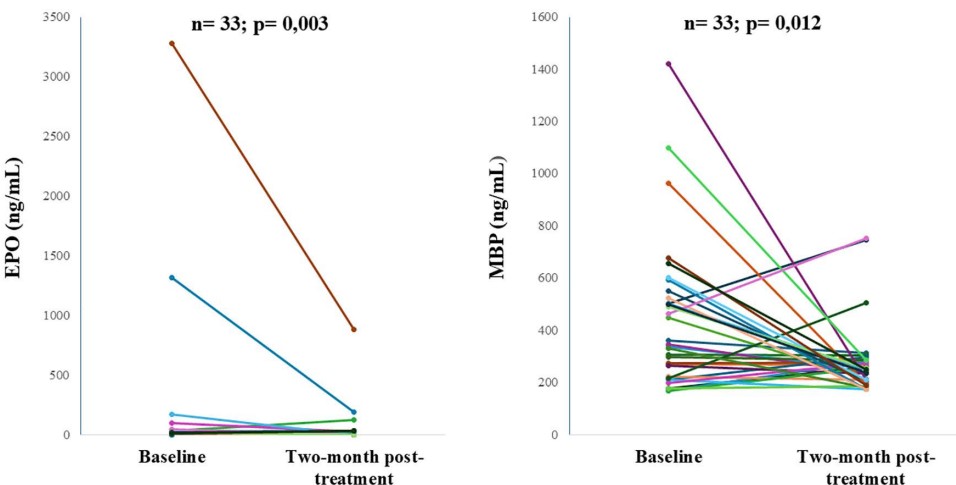

**Fig 3. Changes in serum EPO and MBP levels before and after treatment with antihistamine in the severe CSU (CSU: Chronic spontaneous urticaria; EPO: Eosinophil Peroxidase; MBP: Major Basic Protein; ng/mL: nanogram/milliliter).**

measuring 1329.23 ng/mL compared to 399.55 ng/mL (P < 0.05). Similarly, EPO concentration was also noted, with the CSU group showing a level of 54.96 ng/mL, while the control group exhibited undetectable levels [13].

Our study showed no association between serum EPO and MBP levels with disease activity score (UAS7) and peripheral blood eosinophil counts (P > 0.05), similar to the results of Khanna et al. [13]. This suggests that although eosinophils are involved in the pathogenesis of CSU, they may not play an essential role in the entire group of severe CSU patients. A retrospective study involving over 1,600 patients with CSU revealed that more than 40% of patients in the eosinopenia group had high disease activity (UAS7 ≥ 28), which is nearly double the percentage in the non-eosinopenia group. The study also indicated that eosinopenia is frequently associated with autoimmune conditions [18].

A review study by Fok et al. demonstrated that high disease activity is a biomarker associated with a poor or no response to sgAH1 treatment. Although no specific cut-off point was established, the study indicated that a higher UAS7 score correlates with a poorer response to sgAH1 [19]. Patients with sgAH1-resistant disease are the focus of clinical trials for new drugs in CSU. Consequently, our study specifically examined the severe CSU group to evaluate the relationship between EPO and MBP levels, their treatment response to sgAH1, and changes in these levels after treatment. The results revealed no significant differences in baseline serum EPO and MBP levels among the complete, good, and poor control groups receiving sgAH1. This indicates that the levels of these two substances are not reliable predictors of treatment response to sgAH1 in chronic spontaneous urticaria (CSU). However, there were 2 cases of severe CSU with very high serum EPO and MBP levels (>1000 ng/mL) that both responded poorly to sgAH1. This indicates that eosinophils are involved in only a few patients with CSU resistant to sgAH1. This may be one of the reasons why inhibitors of eosinophil proliferation and chemotaxis, such as IL-5 receptor blocker (Benralizumab), failed in a phase IIb clinical trial for sgAH1-refractory CSU [20]. Both serum EPO and MBP levels tended to decrease after treatment (P < 0.05). SgAH1 interacts with the histamine H1 receptor by keeping it inactive, unlike histamine, which keeps the H1 receptor active. This receptor is present mainly in nerve cells (central and peripheral) and other immune cells such as eosinophils, basophils, etc. [21]. Histamine acts on the H1 receptor on the eosinophil membrane, causing this cell to increase peroxidase synthesis [22]. Therefore, when using sgAH1, eosinophils no longer receive the signal to increase peroxidase synthesis from the H1 receptor, causing a decrease in the concentration of EPO in the serum. The question remains unresolved regarding whether serum MBP levels decrease after administering bilastine-sgAH1. This substance is first synthesized in

a precursor form and then undergoes hydrolysis during granule packaging, forming mature granules in the cytoplasm of eosinophils. The precursor form is highly acidic and rich in glutamic acid, which protects eosinophils from damage as they migrate from the Golgi apparatus to the secondary granules. Notably, the precursor form is absent in mature eosinophils. Although the exact mechanism by which the MBP precursor is hydrolyzed to form MBP is unclear, it may involve IL-5 [23,24]. A previous study has indicated that ketotifen, a first-generation antihistamine, can reduce serum MBP levels in patients with atopic dermatitis [25]. Subsequent research revealed that ketotifen inhibits eosinophil chemotaxis at inflammation sites during allergic reactions [26]. This inhibition indirectly reduces eosinophil degranulation, lowering the serum's MBP levels. A sgAH1, cetirizine, has also been shown to inhibit eosinophil chemotaxis induced by chemokines such as platelet-activating factor and N-formyl methionyl leucyl phenyl alanyl [27]. More studies are needed to determine whether bilastine has the same inhibitory effect on eosinophil chemotaxis as ketotifen and cetirizine.

The main limitation of our study is that we did not quantify the concentration of EPO and MBP in the skin lesions of CSU patients and compare them with the concentration in the serum. This gives us no general view of the role of EPO and MBP concentration in CSU. Moreover, this investigation was conducted at a single center and exclusively involved Vietnamese subjects. As a result, there are inherent limitations in extrapolating the findings to the broader population of individuals with CSU.

## Conclusion

In conclusion, serum EPO concentrations were higher in the CSU group than in the healthy controls but were not associated with disease activity. Serum MBP concentrations did not differ between the CSU group and the healthy controls. Neither EPO nor MBP concentrations were biomarkers of response to antihistamine treatment, and both decreased after treatment with this drug. Eosinophils are not the primary effector cells in CSU. Additional research is necessary to clarify the complex pathogenesis of this condition.

## Supporting information

**S1 Table. Characteristics of the study subjects.** (ASST: autologous serum skin test; CRP: C-reactive protein; CSU: chronic spontaneous urticaria; IgE: immunoglobulin E; IgG: immunoglobulin G; TPO: Thyroid Peroxidase; UAS7: urticaria activity score over 7 days; *Man-Whitney U test,** $\chi^2$ test, ***Fisher Exact test, +T-test).
(DOCX)

**S2 Table. Serum EPO and MBP concentrations in CSU according to some clinical/paraclinical characteristics.** (ASST: autologous serum skin test; CRP: C-reactive protein; CSU: chronic spontaneous urticaria; EPO: Eosinophil Peroxidase; IgE: immunoglobulin E; IgG: immunoglobulin G; MBP: Major Basic Protein; TPO: Thyroid Peroxidase; *Mann-Whitney U test, ++Kruskal-Wallis test).
(DOCX)

## Acknowledgments

Assistance with the study: None

## Author contributions

**Conceptualization:** Cuc Thi Kim Nguyen, Minh Nguyet Vu, Lan Thi Pham.

**Data curation:** Cuc Thi Kim Nguyen, Minh Nguyet Vu.

**Formal analysis:** Cuc Thi Kim Nguyen, Minh Nguyet Vu.

**Funding acquisition:** Cuc Thi Kim Nguyen.

**Investigation:** Cuc Thi Kim Nguyen, Minh Nguyet Vu, My Huyen Le.

**Methodology:** Cuc Thi Kim Nguyen, Minh Nguyet Vu, Lan Thi Pham, My Huyen Le, Doanh Huu Le.

**Project administration:** Cuc Thi Kim Nguyen, Minh Nguyet Vu, Doanh Huu Le.

**Resources:** Cuc Thi Kim Nguyen.

**Software:** Cuc Thi Kim Nguyen, Minh Nguyet Vu.

**Supervision:** Cuc Thi Kim Nguyen, Minh Nguyet Vu, Lan Thi Pham, My Huyen Le.

**Validation:** Cuc Thi Kim Nguyen, My Huyen Le, Doanh Huu Le.

**Visualization:** Cuc Thi Kim Nguyen, Minh Nguyet Vu.

**Writing – original draft:** Cuc Thi Kim Nguyen, Minh Nguyet Vu, Lan Thi Pham, My Huyen Le, Doanh Huu Le.

**Writing – review & editing:** Cuc Thi Kim Nguyen, Minh Nguyet Vu, Lan Thi Pham, My Huyen Le, Doanh Huu Le.

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
