## [Decision Letter · Decision Letter 0]

30 Sep 2025

Dear Dr. Vu,

We look forward to receiving your revised manuscript.

Kind regards,

Ping Xiang

Academic Editor

PLOS ONE

2. Please include a separate caption for each figure in your manuscript.

3. We notice that your supplementary figures are uploaded with the file type 'Figure'. Please amend the file type to 'Supporting Information'. Please ensure that each Supporting Information file has a legend listed in the manuscript after the references list.

4. We notice that your supplementary tables are included in the manuscript file. Please remove them and upload them with the file type 'Supporting Information'. Please ensure that each Supporting Information file has a legend listed in the manuscript after the references list.

Reviewers' comments:

Reviewer's Responses to Questions

**Comments to the Author**

1. Is the manuscript technically sound, and do the data support the conclusions?

Reviewer #1: Yes

Reviewer #2: Yes

Reviewer #3: Partly

2. Has the statistical analysis been performed appropriately and rigorously?

Reviewer #1: I Don't Know

Reviewer #2: Yes

Reviewer #3: Yes

3. Have the authors made all data underlying the findings in their manuscript fully available?

Reviewer #1: Yes

Reviewer #2: Yes

Reviewer #3: No

4. Is the manuscript presented in an intelligible fashion and written in standard English?

Reviewer #1: Yes

Reviewer #2: No

Reviewer #3: Yes

Reviewer #1: This manuscript investigates eosinophil peroxidase and major basic protein levels in patients with chronic spontaneous urticaria (CSU), through across-sectional study con 150 participant .

The manuscript are well-structured and performed. However, normality testing should be reported when evaluating the relationship between serum levels and UAS7, with the corresponding correlation method applied accordingly. The same requirement applies to comparisons of EPO/MBP mean values between groups (e.g., controls vs. CSU).

The discussion aligns with the observed results and is supported by the current body of evidence in the literature.

Reviewer #2: The findings of the present article shows significance of serum Eosinophil

14 Peroxidase (EPO) and Major Basic Protein (MBP) levels in chronic spontaneous urticaria (CSU).

The investigators have highlighted the significance of EPO and MBP in CSU in the patient data and have selected proper control to show their finding. The experiments results and findings are appropriately described.

However, the manuscript needs few minor revision, which are as follows:

1. The title of the manuscript is not impressive and requires rephrasing to make it more relevant to the study.

2. The English language is required to be improved in whole article.

3. Graph of the figures need to be improved. the lines of the y-axis and x-axis are not visible.

4. The investigators should also describe briefly about measurement of EPO and MBP levels by ELISA kit.

Reviewer #3: In this manuscript entitled “Serum Eosinophil Peroxidase and Major Basic Protein Levels in Chronic Spontaneous Urticaria”, the authors aim to evaluate whether serum eosinophil peroxidase (EPO) and major basic protein (MBP) levels correlate with disease severity or antihistamine response in chronic spontaneous urticaria (CSU). This reviewer believes that the topic may be beneficial for readers of PLOS ONE to some extent. However, the manuscript requires significant revisions before it can be considered for publication.

Comments and Concerns:

1. In the manuscript, the study was conducted in a single center with a homogeneous population and a cohort of 150 subjects (60 severe CSU, 60 non-severe CSU, and 30 healthy controls). This design limits the impact and generalizability of the findings. The authors should explicitly acknowledge this limitation to help readers interpret the applicability of the results to broader CSU populations.

2. The manuscript focuses exclusively on two eosinophil-related biomarkers, which makes the scope somewhat narrow. The authors should better contextualize the limited clinical utility of EPO and MBP and discuss other relevant inflammatory markers or pathways to provide a broader perspective on CSU pathogenesis.

3. The analysis centers on serum EPO and MBP levels, but eosinophil activity in CSU may be more accurately reflected in lesional skin rather than in circulation. The absence of lesional skin data weakens the strength of the claims. At minimum, this limitation should be highlighted and discussed.

4. While the statistical analyses are adequate, more in-depth approaches (e.g., multivariate modeling) could be applied to strengthen the conclusions. Furthermore, the authors should be encouraged to report effect sizes with 95% confidence intervals, not only p-values, to improve interpretability and clinical relevance.

5. The figures are low resolution and not presented professionally, and the figure order appears inconsistent. Additionally, tables and figures should be included in the main text rather than as Supporting information (e.g., not as “S1_Table 1” or “S3_Fig.1”). Improving figure quality and integrating them properly into the manuscript would significantly enhance readability.

**Do you want your identity to be public for this peer review?** For information about this choice, including consent withdrawal, please see our Privacy Policy

Reviewer #1: No

Reviewer #2: No

Reviewer #3: **Yes: ** Guanghui Han

---

## [Author Response · Author response to Decision Letter 1]

2 Oct 2025

Response to Reviewers

Reviewer #1: This manuscript investigates eosinophil peroxidase and major basic protein levels in patients with chronic spontaneous urticaria (CSU) through a cross-sectional study of 150 participants.

The manuscript is well-structured and well-performed. However, normality testing should be reported when evaluating the relationship between serum levels and UAS7, with the corresponding correlation method applied accordingly. The same requirement applies to comparisons of EPO/MBP mean values between groups (e.g., controls vs. CSU).

The discussion aligns with the observed results and is supported by the current body of evidence in the literature.

Response: We appreciate your feedback. We have incorporated the normal distribution test into the article.

Reviewer #2: The findings of the present article show the significance of serum Eosinophil Peroxidase (EPO) and Major Basic Protein (MBP) levels in chronic spontaneous urticaria (CSU).

The investigators have highlighted the significance of EPO and MBP in CSU in the patient data and have selected a proper control to show their findings. The experimental results and findings are appropriately described.

However, the manuscript needs a few minor revisions, which are as follows:

1. The title of the manuscript is not impressive and requires rephrasing to make it more relevant to the study.

Response: We appreciate your valuable feedback. We will revise the title of the article to "The Relationship between Serum Eosinophil Peroxidase and Major Basic Protein Levels in Relation to Severity and Response to H1 Antihistamines in Chronic Spontaneous Urticaria."

2. The English language is required to be improved throughout the whole article.

Response: We sincerely appreciate your valuable feedback. We have made revisions to the language used in the article to enhance its professionalism and align it with academic standards.

3. The graph of the figures needs to be improved. the lines of the y-axis and x-axis are not visible.

Response: We appreciate your valuable feedback. We have revised the chart according to your comments.

4. The investigators should also describe briefly the measurement of EPO and MBP levels by the ELISA kit.

Response: We appreciate your valuable feedback. We have revised the manuscript according to your comments.

Reviewer #3: In this manuscript entitled “Serum Eosinophil Peroxidase and Major Basic Protein Levels in Chronic Spontaneous Urticaria”, the authors aim to evaluate whether serum eosinophil peroxidase (EPO) and major basic protein (MBP) levels correlate with disease severity or antihistamine response in chronic spontaneous urticaria (CSU). This reviewer believes that the topic may be beneficial for readers of PLOS ONE to some extent. However, the manuscript requires significant revisions before it can be considered for publication.

Comments and Concerns:

1. In the manuscript, the study was conducted in a single center with a homogeneous population and a cohort of 150 subjects (60 severe CSU, 60 non-severe CSU, and 30 healthy controls). This design limits the impact and generalizability of the findings. The authors should explicitly acknowledge this limitation to help readers interpret the applicability of the results to broader CSU populations.

Response: We appreciate your valuable feedback and have noted it in the limitations section of the study.

2. The manuscript focuses exclusively on two eosinophil-related biomarkers, which makes the scope somewhat narrow. The authors should better contextualize the limited clinical utility of EPO and MBP and discuss other relevant inflammatory markers or pathways to provide a broader perspective on CSU pathogenesis.

Response: We sincerely appreciate your valuable feedback. In our investigation of biomarkers associated with CSU, we conducted an assessment of the correlation between serum EPO and MBP concentrations and the laboratory indices recommended by the European Academy of Allergy and Clinical Immunology (EAACI). These indices encompass blood counts, with a particular emphasis on eosinophil and basophil counts, as well as C-reactive protein (CRP) concentrations, total immunoglobulin E (IgE), and anti-thyroid peroxidase (anti-TPO) IgG levels. We noted, however, that other specialized biomarkers for CSU, including IgE autoantibodies related to autoimmune urticaria type 1 (such as anti-TPO IgE and anti-EPO IgE) and anti-FcɛRI or anti-IgE IgG autoantibodies, along with the basophil histamine release test for autoimmune urticaria type 2b, are not routinely available at urticaria centers. Consequently, we were unable to evaluate their relationships in our current study.

3. The analysis centers on serum EPO and MBP levels, but eosinophil activity in CSU may be more accurately reflected in lesional skin rather than in circulation. The absence of lesional skin data weakens the strength of the claims. At a minimum, this limitation should be highlighted and discussed.

Response: We appreciate your feedback and have noted it in the limitations section of the study.

4. While the statistical analyses are adequate, more in-depth approaches (e.g., multivariate modeling) could be applied to strengthen the conclusions. Furthermore, the authors should be encouraged to report effect sizes with 95% confidence intervals, not only p-values, to improve interpretability and clinical relevance.

Response: We appreciate your valuable feedback. Serum EPO and MBP concentrations demonstrated no correlation with UAS7 and UCT scores. Upon plotting the ROC curve to identify the threshold for serum EPO and MBP concentrations that would effectively differentiate between severe and non-severe CSU, as well as H1-responsive and non-responsive CSU, it was determined that neither of these threshold values achieved statistical significance regarding the area under the curve. Consequently, these threshold values were not incorporated into the multivariate regression model evaluating risk factors for severe CSU and uncontrolled CSU. Therefore, this section has been excluded from the manuscript.

5. The figures are low resolution and not presented professionally, and the figure order appears inconsistent. Additionally, tables and figures should be included in the main text rather than as Supporting information (e.g., not as “S1_Table 1” or “S3_Fig.1”). Improving figure quality and integrating them properly into the manuscript would significantly enhance readability.

Response: We appreciate your valuable feedback. We have revised the manuscript according to your comments.

---

## [Decision Letter · Decision Letter 1]

22 Oct 2025

The Relationship between Serum Eosinophil Peroxidase and Major Basic Protein Levels in Relation to Severity and Response to H1 Antihistamines in Chronic Spontaneous Urticaria

PONE-D-25-38673R1

Dear Dr. Minh,

We’re pleased to inform you that your manuscript has been judged scientifically suitable for publication and will be formally accepted for publication once it meets all outstanding technical requirements.

Kind regards,

Ping Xiang

Academic Editor

PLOS ONE

Additional Editor Comments (optional):

Reviewers' comments:

Reviewer's Responses to Questions

**Comments to the Author**

Reviewer #1: All comments have been addressed

Reviewer #2: All comments have been addressed

Reviewer #3: All comments have been addressed

2. Is the manuscript technically sound, and do the data support the conclusions?

Reviewer #1: Yes

Reviewer #2: Yes

Reviewer #3: Yes

3. Has the statistical analysis been performed appropriately and rigorously?

Reviewer #1: Yes

Reviewer #2: (No Response)

Reviewer #3: Yes

4. Have the authors made all data underlying the findings in their manuscript fully available?

Reviewer #1: Yes

Reviewer #2: Yes

Reviewer #3: Yes

5. Is the manuscript presented in an intelligible fashion and written in standard English?

Reviewer #1: Yes

Reviewer #2: Yes

Reviewer #3: Yes

Reviewer #1: The revised version of the manuscript is appreciated, as it satisfactorily incorporates the modifications suggested by the reviewers. The resulting article is well structured and presents a coherent and rigorous analysis of the findings. The conclusions are well aligned with the study design and appropriately contextualized within the scope of the cohort investigated. Overall, the revised manuscript meets the essential criteria of clarity, methodological soundness, and scientific relevance required for consideration in publication.

Reviewer #2: (No Response)

Reviewer #3: (No Response)

**Do you want your identity to be public for this peer review?** For information about this choice, including consent withdrawal, please see our Privacy Policy

Reviewer #1: No

Reviewer #2: No

Reviewer #3: No

---

## [Editor Report · Acceptance letter]

PONE-D-25-38673R1

PLOS ONE

Dear Dr. Vu,

I'm pleased to inform you that your manuscript has been deemed suitable for publication in PLOS ONE. Congratulations! Your manuscript is now being handed over to our production team.

Kind regards,

on behalf of

Professor Ping Xiang

Academic Editor

PLOS ONE